Manuscript prepared for Hydrol. Earth Syst. Sci.
with version 2015/09/17 7.94 Copernicus papers of the LATEX class copernicus.cls.
Date: 7 April 2016

# Geoscience on television: a review of science communication literature in the context of geosciences

Rolf Hut[1], Anne M. Land-Zandstra[2], Ionica Smeets[2], and Cathelijne R. Stoof[3]

[1]Delft University of Technology, Faculty of Civil Engineering and Geosciences, chair of Water Resources Engineering
[2]Science, Communication and Society, Leiden University
[3]Soil Geography and Landscape Group, Wageningen University

*Correspondence to:* Rolf Hut (r.w.hut@tudelft.nl)

**Abstract.** Geoscience communication is becoming increasingly important as climate change increases the occurrence of natural hazards around the world. Few geoscientists are trained in effective science communication, and awareness of the formal science communication literature is also low. This can be challenging when interacting with journalists on a powerful medium like TV. To provide geoscience communicators with background knowledge on effective science communication on television, we reviewed relevant theory in the context of geosciences and discuss six major themes: scientist motivation, target audience, narratives and storytelling, jargon and information transfer, relationship between scientists and journalists, and stereotypes of scientists on TV. We illustrate each theme with a case study of geosciences on TV and discuss relevant science communication literature. We then highlight how this literature applies to the geosciences and identify knowledge gaps related to science communication in the geosciences. As TV offers a unique opportunity to reach many viewers, we hope this review can not only positively contribute to effective geoscience communication but also to the wider geoscience debate in society.

## 1 Introduction

As climate change increases the occurrence of natural hazards in many parts of the world (Batllori et al., 2013; Gobiet et al., 2014; Hirabayashi et al., 2013), accurate and effective communication of hydrogeoscience information to the general public is becoming increasingly important. Yet, few researchers are trained in effective science communication (Besley and Tanner, 2011), and science communication research is typically not standard fare for geoscientists (Stewart and Nield, 2013). There is a wealth of literature on effective science communication in general (e.g., Bucchi and Trench, 2014; WEIGOLD, 2001), though there is little on the geosciences in particular (e.g., Forster and Freeborough, 2006; Stewart and Nield, 2013; Liverman, 2008).

More theoretical knowledge on science communication can provide useful background knowledge for geoscience communicators, and thereby positively contribute to the geoscience debate in

society. With this latter goal in mind, in this paper, we provide geoscientists with a review of relevant general science communication theory. We focus on television appearances of (geo)scientists, though much of the research/themes/literature discussed also holds for popular-scientific presentations or for interactions with newspaper journalists. We use the term television loosely, also applying it to programs that appear online.

Television is by definition a form of one-way communication, and in this context effective science communication is aimed at popularizing science and enhancing the scientific literacy of the general public. However, within the field of science communication this top-down-approach has been criticized (Sturgis and Allum, 2004). Two-way communication is important, for instance since scientific literacy is not the only factor in how people perceive climate change and other risks (Kahan et al., 2012). In hazard situations it is especially important to engage the public instead of just informing them (Frewer, 2004). However, this will be easier if there is a broader "geo-literacy" among the general public (Stewart and Nield, 2013) and despite its limitations one-way communication has its own merits (Sturgis and Allum, 2004),(Wright, 2006). Recent results also show that the general public values one-way communication by experts, indicating that the public is looking for new knowledge and understanding of science (Fogg-Rogers et al., 2015).

In this review we discuss six major themes in science communication research related to television performances: scientist motivation (Section 2), target audience (Section 3), jargon and information transfer (Section 4), narratives and storytelling (Section 5), relationship between scientists and journalists (Section 6), and stereotypes of scientists (Section 7). For each theme we make the results from the literature tangible by analyzing a television appearance of a geoscientist from a science communication perspective. For these case studies we use examples for which we had background information on behind-the-scenes discussions and negotiations, namely television appearances of authors Hut and Stoof.

## 2 Motivation

Most scientists consider it part of their duty to interact with the media (Dijkstra et al., 2015; Peters, 2013; Liverman and Jaramillo, 2011): they feel a responsibility to inform the public about their field of research and many regularly respond to journalists. Peters (2013) analyzed multiple surveys assessing the relationship between scientists and the media and found that in general over 60% of scientists had been in contact with journalists in the previous three years. However, a sense of responsibility is not the only reason that scientists can have for their media presence. In the next example, Hut explains why he responded to a request to feature in a Dutch science talkshow.

**2.1   Case: 'De Wereld Leert Door' science talkshow**

The science-focused daily talkshow 'De Wereld Leert Door'[1] is a spin-off of the popular Dutch prime-time talkshow 'De Wereld Draait Door'[2]. In the 13-min show, a scientist is interviewed about their work. While scientists are usually invited much ahead of time, Hut was invited on very short notice (8 hours before the show was to be taped) to replace a sick colleague (Verening Arbeiders Radio Amateurs, 2013). Given the time crunch, the producers were under some stress to 'book' a scientist for the episode that day. This gave Hut some leeway in the determining the subject of the interview. Hut reasoned that by discussing how current research is incorporated into teaching, high school kids (potential students) may be triggered to go to Delft University of Technology for their BSc and MSc education. Therefore Hut wanted the interview to focus on the way he uses his research (developing geoscientific sensors from simple electronics found in any home, kitchen or garage, also called 'tinkering') in his teaching, both in terms of the classes he teaches and of his students' projects. In the first request for participation, Hut stated that he would like to discuss (and bring physical demonstrations of) the sensors made by his students. The producers agreed to this, considerably shifting the intended focus of the interview away from Hut's own research. The resulting interview shortly discussed Hut's own research and then gradually changed topic to his classes, in which students develop their own sensors without being allowed to spend any money. By entering a dialogue with the journalists before agreeing to participate Hut ensured that not only the journalist, but also he could get out of the interview what he wanted, namely highlighting his teaching. An unintentional consequence of Hut's appearance in 'De Wereld Leert Door' was that he became the 'go-to-person' for some journalists on the topic of 'scientific tinkering'. This may have helped in landing Hut a position as column writer for a major Dutch newspaper.

**2.2   Relevant science communication literature and reflection**

The science talk show in which Hut featured was intended to inform the public about science, which is another motivation for scientists to interact with the media (Nield, 2008). Peters (2013) reported that over 90% of biomedical researchers in five different countries list a better educated public as an important reason for science communication. Television is a strong medium in this light, as it is the main source of science information for the general public (Stewart and Nield, 2013). Miller et al. (2006) found that science communication on TV can be very effective. They studied how well adult viewers in the United States remembered seeing science stories in local television news shows, and found that on average around 60% of potential viewers remembered seeing one or more of the science stories that were shown on their local news. Of the people who remembered seeing the stories, around 44% could give an explanation of the science content. This may not seem much, but this resulted in a total number of 4.1 million people recalling the stories and 1.8 million being able to

---

[1]The world keeps on learning

[2]The world keeps on turning

explain the science. Another important reason for scientists to engage with the media highlighted by Peters (2013) is the need for government funding, as receiving government funding (from taxpayer money) comes with the accountability towards the taxpayers who have a right to know what has been done with their money. Meanwhile, media attention is assumed to lead to more public goodwill, which in turn may lead to more citations and more funding (Phillips et al., 1991; Weingart, 1998). Or in Hut's case, also more prospective students.

Besides these overarching motivations for researchers to interact with the media, Hut's science talkshow example shows that other motivations can become important in specific situations. First, because of the short deadline, Hut hoped to have greater control over the content of his performance on the show, allowing him to also talk about his teaching. He thereby realized the opportunity that came with the time crunch as it did indeed increase his degree of control over the content, which is often an issue in science communication. This is also clear from Peters (2013) review, in which lack of control over the communication process (including content) was identified as one of the barriers for scientists dealing with the media. This is further discussed in Section 6. Secondly, Hut also concluded that this performance made him the "go-to person" for some journalists on tinkering for science which likely helped in landing him a gig as regular column-writer for a national newspaper. So although this was initially not a reason for him to participate in the show, media exposure can help scientists expand their science communication activities, and in many cases even have a snowball-effect (Albaek, 2011).

In conclusion, important reasons for scientists to interact with the media include a sense of responsibility, wanting to educate the public, showing accountability towards taxpayers, potentially increasing scientific citations and research funding. In addition, appearing on television in particular may be an effective way to reach a broad audience and may increase visibility of scientists for journalists, leading to increased exposure.

## 3   Target audience

In any form of (science) communication it is important to think about who it is you are talking to, your target audience (Liverman, 2008). What are the education levels of viewers or listeners? What is their prior knowledge of a certain topic? But also, why are they watching? What are their expectations? For example, differences between adults and children result in different approaches on television. Programs for children often have the goal to educate and tend to be produced in series, while adult programming is more often focused on a single topic in a given show (Feder et al., 2009). There are various distinct genres of science on television: news, entertainment, fiction, nonfiction, documentary, magazine style (Dhingra, 2003). Each genre comes with its own target audience, as in-depth documentaries will have a different target audience than science stories during news shows. Because different target audiences require different communication approaches, it is extremely dif-

ficult - if not impossible - to successfully reach more than one target audience; something Hut found out the hard way.

### 3.1 Case: 'Universiteit van Nederland' public lecture: 'Waarom hoeft wetenschappelijk onderzoek helemaal niet duur te zijn'

Despite its name, De Universiteit van Nederland[3] is not an institute of higher education but a non-profit that invites lecturers from Dutch universities to give a set of five short public lectures. These 15-min lectures are recorded for live audience in a nightclub in Amsterdam, The Netherlands, professionally edited and posted on YouTube (Universiteit van Nederland, 2014b). Hut was asked to record a series of lectures, one of which was called 'Waarom hoeft wetenschappelijk onderzoek helemaal

niet duur te zijn'[4]. Since Hut would receive no financial compensation for his considerable time investment required, his employer only agreed to Hut's participation when the public lectures would also be useful in Hut's academic teaching. Hut's solution was to use the public lectures in his classes in which he 'flipped' the classroom, by assigning the videos as homework and using class time for discussion and problem sets to stimulate higher order thinking skills. This solution created two con-

trasting target audiences: the general television audience with no relevant scientific background and Hut's third-year civil engineering students. During rehearsal it became clear that catering to both target audiences at the same time would be impossible: the mathematics Hut included for his students were not suited for the general television audience. Likely because of that, almost all mathematics was cut from the final online video by UvNL editors, which is visible in sudden changes on the

blackboard (Universiteit van Nederland (2014b) at 13:50 min). To still be able to use the videos in his classes, Hut ended up editing the raw camera feeds to create teaching-appropriate versions of the lectures that he uploaded on his own YouTube channel. This is a clear example that one is unlikely to reach multiple distinct audiences with a single lecture.

### 3.2 Relevant science communication literature and reflection

The target audience for UvNL is a general audience without specific scientific background. In a national newspaper article the founder of UvNL, Alexander Klöpping, and the editor-in-chief, Frouke van Goethem, explain that the goal of the program is to spark excitement for science among an audience of laypeople (Weijer, 2014). Klöpping explains that when posted on YouTube, their videos will compete with cat videos, "so it needs to look good and show some of the excitement in the room".

Van Goethem tries to prepare lecturers for this by asking them to pretend that they are at a birthday party telling people what they are doing. That is the level of complexity, or simplicity, that is aimed for in the program, preferably without complicated formulas on the blackboard. This illustrates how the two target audiences that Hut wanted to reach (the general audience as well as his own students)

---

[3]University of the Netherlands
[4]Why scientific research does not have to be expensive

were irreconcilable. For the general audience, the lectures had to include as little complicated information as possible, while the mathematical background of the concepts explained was essential for the third-year bachelor students. This resulted in videos with weird cuts and a lot of extra work for Hut to make the material appropriate for his teaching.

Another common problem in terms of target audience is that many scientists consider their peers as their main audience when they appear in the media (Dijkstra et al., 2015; Dunwoody, 2012; Peters, 2013): scientists tend to be more concerned about the judgment of their colleagues than about the response of the general audience. Peters (2013) described a survey among biomedical scientists in Germany and the United States, in which they were asked which conditions they thought were important to their peers before they or other scientists appeared in the media: scientists should be of excellent scientific reputation and have broad experience on the topic in question; stick to the facts and avoid going beyond the facts; and avoid putting the spotlight on themselves. Moreover, results should have already been published in scientific journals, and the broadcast medium must have shown high-quality science reporting. Fulfilling these conditions very likely interferes with successfully reaching the general audience, which is the primary audience of most media outlets.

To sum up, when engaging with the media it is essential to know the target audience and to adjust the communication approaches accordingly. It is a major challenge, and highly unlikely, to effectively cater to more than one target audience. This also means that delivering a successful media performance is difficult when concerned about the judgement of other scientists. Whether this perceived peer pressure is justified or not remains to be studied, not just in the geosciences but also in the general field of science communication. One aspect of catering to a specific target audience that has been studied is to adapt the language that is used, by avoiding jargon and using storytelling techniques; the topics of the next two sections.

## 4 Jargon and information transfer

It is very easy to overestimate the literacy of your audience. The Organisation for Economic Co-operation and Development (OECD) classifies six levels of literacy on a 500-point scale (Outlook, OECD Skills, 2013). At the two lowest levels (level 1a and 1b), people only understand short texts with basic vocabulary. Only at the highest level (level 5), people understand texts that use specialised background knowledge, and a meager 0.7% of the population achieves this level. In a worldwide survey of population literacy, the average score was 273 (level 2), which is comparable to a reading level of an 11-year old child. Even in developed countries like The Netherlands, 40% of the adults score at level 2 or below. This is comparable to level B1 on the Common European Framework of Reference for Languages (CEFR) scale, which means that most people can not understand technical terms (Council of Europe, 2000). When communicating science, whether in writing or on television, this relatively low level of literacy of your potential audience and the challenges they may face in

understanding technical terms or jargon is important to bear in mind. In light of this, we discuss a
case where Stoof was interviewed and in her opinion did not use any jargon.

### 4.1 Case: 'Netwerk' documentary of field visit

On 28 Aug 2009, a 155-ha wildfire in the dunes along the Dutch coast threatened the village of
Schoorl. This (for Dutch standards) very large fire caused evacuations of 550 people, including 200
homes and an elderly home. Four days after the fire, Stoof was invited to visit the burned area with the
forest manager and a camera crew for an 8-min TV documentary for Netwerk, a daily news program
broadcast on national TV immediately after the 8 o'clock evening news (Nederlandse Christelijke
Radio-Vereniging, 2009). The field visit was informal, and most of the content was filmed during a
hike through the burned area. While talking to the forest manager, Stoof asks him for clarification
of burned patterns, fire behavior and suppression activities, and discusses fire impact on the soil,
signs of water erosion caused by fire services hosing down the fire, and potential impact of the fire
on water and wind erosion. They visit the area while fire services are still searching for smouldering
hotspots in the litter layer. At some point, they find a hotspot before fire services are present and
Stoof starts to dig a trench in the organic material to stop the fire with her bare hands.

### 4.2 Relevant science communication literature and reflection

The use and impact of jargon has been mostly and extensively studied in the medical sciences, and
particularly in patient-doctor-interactions (Boyle, 1970; Hadlow and Pitts, 1991; Blackman and Sa-
hebjalal, 2014). As there is no specific definition of jargon in the geosciences, we adopt the standard
definition from medical sciences (Castro et al., 2007) where jargon is defined as 1) technical terms
with only one meaning listed in a technical dictionary, and 2) terms with a different meaning in
lay contexts. The Netwerk documentary contains both types of jargon. For instance, Stoof uses the
following technical terms: water erosie (water erosion), wind erosie (wind erosion), minerale bodem
(mineral soil), humuslaag (humus layer), vegetatie (vegetation) and suppressie (suppression, which
in correct Dutch translation would actually be 'repressie').The forest manager refers to: monotonous
pine forest and fire lane. He also uses some terms that have a different meaning in lay contexts, such
as 'poor soil'. It may be an eye opener to those ingrained in the geosciences, but most viewers will
not immediately understand what it means for soil to be poor. For people with limited literacy, i.e.
a large part of the viewers, these technical and ambiguous words are confusing. This does not mean
that all jargon on television should be avoided, but care should be taken to explain these terms. Stoof
does this nicely when she talks about water erosion and in one sentence explains that this is caused
by rainfall that is not absorbed by the soil

In the geosciences, many words that may be strictly defined within a specific sub field of geo-
sciences have a more general meaning to lay people, such as the difference between a brook and
a stream, or that a river is also a stream. Therefore, geoscientists should be particularly aware of

the second type of jargon, especially when communicating about the nuances of probability and uncertainty relating to geoscientific forecasts. In the geosciences, probabilistic forecasts are often expressed in return periods (i.e. a once in a hundred year flood), which are often misinterpreted by those not statistically educated (Boykoff, 2008; Ward, 2008);

For researchers it might seem that it is impossible to communicate without using jargon. But it is worthwhile to realise that information with a relatively small scientific depth can already have an enormous impact on laymen. Robbert Dijkgraaf, former president of the Dutch Royal Society for Arts and Sciences, suggested that we should measure the information transfer in science communication as the product of information load and the size of the audience (Dijkgraaf, 2006). This is especially important to remember with regards to television which is a true mass medium. Stoof's short sentence about water erosion reached 579.000 laymen viewers and in a sense transferred more information than a detailed conference talk attended by a hundred experts. Using accessible language on TV is paramount as viewers are easily scared away by things they do not understand. In Land-Zandstra and De Bakker (2014) Henk Bas, editor-in-chief for numerous Dutch popular-scientific television programmes, says that viewers report in questionnaires that they would like more substance in science shows. But when their concentration levels are measured, viewers' attention drops during the segments where scientists explain things. Bas concludes that content has to be presented in an amusing way to work on television.

Summarizing, jargon is much broader than many scientists may think. The extent to which geoscientists use jargon in laymen communications, and what exactly constitutes jargon in the geosciences has not been studied, but more knowledge on this topic could help geoscientists to tailor their science communication activities more effectively to their intended audience. Finally, it is important to remember that television offers a unique opportunity to reach hundred thousands of viewers and that explaining even one small detail to such a huge audience is already a big result.

## 5 Narratives and storytelling

In academic interactions, whether on paper or at conferences or in seminars, conclusions are usually presented at the end (Fig. 2, right pyramid). This is in stark contrast to journalism where it is common to start with the conclusions ( Fig. 2, left pyramid). This latter structure is called the inverted pyramid (Po ttker, 2003): the most important information comes first and the lead sentence answers the five w-questions: who? when? where? what? why? The rest of the facts are then presented in decreasing order of importance. Stewart and Nield argue that geoscientists can communicate more effectively with laymen when they invert the pyramid of their presentations (Stewart and Nield, 2013). The paper that first introduced the inverted pyramid (Somerville and Hassol, 2011) to geoscientists uses a down pointing triangle for the scientists communicating with peers and an up pointing triangle for scientists communicating with lay people to convey the message that "the scientific method trains

scientists to begin with outlining the general academic context and then work through to some narrow research finding, which they then proceed to communicate. The starting point for the public, on the other hand, is the sharp question of 'so what?', and addressing that successfully can take them deeper into the academic detail and wider context." (Stewart, 2016). Here, we use the convention used by journalists, where the inverted pyramid of communicating with lay people is a down pointing triangle. We recognize that this is a matter of personal preference and at the same time, that the different conventions used in different fields to communicatie ideas like the inverted pyramid is a valid subject of academic study in its own right.

In recent years there has been a paradigm shift in television journalism, from the inverted pyramid (Fig. 2, left pyramid) to narrative structures that use storylines similar to those in fiction (Ytreberg, 2001). As Downs (2014) describes it: "Narrative can captivate the audience, driving anticipation for plot resolution, thus becoming a self-motivating vehicle for information delivery."

In the following example Hut used a narrative tool to bring his research to a larger audience.

## 5.1 Case: 'Universiteit van Nederland' public lecture: 'Waarom varen boten nog steeds tegen bruggen aan?'

In the 'Waarom varen boten nog steeds tegen bruggen aan?'[5] lecture for the 'University of the Netherlands' (Section 3.1), using the example/application of measuring whether a boat fits under a bridge was chosen to 'anchor' the entire lecture (Universiteit van Nederland, 2014a). The lecture subject, mathematical sampling theory, answers the question how often one needs to sample to adequately measure a given phenomenon without errors. This can be quite abstract and Hut chose to continuously refer back to the central question: how can we be sure that a boat fits under a closed bridge? Hut intentionally did this to explain why the theoretical material he discussed is important. He starts his talk by asking his audience why boats keep hitting bridges and explains the magnitude of the disaster when collisions were to occur. He tells how people historically measured whether a boat would fit underneath a bridge and why this method does not always work. Then he delves into tides, and how waves with different wavelengths influence the height of a ship differently. His story becomes rather technical, but because the audience wants the solution to the initial question they are drawn into the story.

## 5.2 Relevant science communication literature and reflection

In his talk Hut uses the basic narrative form of complication-resolution that has been well-tested in journalism. Franklin (1994) describes and analyses how he used this technique in his two Pulitzer prize-winning non-fiction stories. Franklin goes much further in using narratology than is probably necessary for geoscientists on TV, but using some of the tools from storytelling can already help to engage the audience, especially when talking about complex or abstract subjects. Hut introduced

---

[5]Why do boats keep hitting bridges?

each of his five talks with a question and we noted their number of YouTube views on 23 Oct 2015, one year after the videos were published (Table 1). Since it is a series, one would expect the most views for the first lecture, but in fact the second and third have been viewed more and we suspect that the attractiveness of the questions in their titles and the need of the audience for their solution is one of the reasons. Dahlstrom (2014) gives a broad overview of the literature on narratives and encourages scientists to use them in their communication. He argues that research "generally suggests that audiences are more willing to accept normative evaluations from narratives than from more logical-scientific arguments". This means that it is more convincing to present one specific case from which the audience can generalize the underlying idea, than to try and prove a point in an abstract way. This even holds when the chosen narrative or storyline is fictional. Where science should be based on evidence and not on anecdotes, science communication is more effective when it uses stories rather than pure evidence (Stewart and Nield, 2013). Another example of storytelling in science communication is following a character over a time period (Dahlstrom, 2014). TV shows often focus on personal stories about the people behind the research to get a narrative that engages the audience.

In short, there are many ways to incorporate narratives in geoscience communication and especially the richness in anecdotes and stories about natural hazards offer excellent starting points. As Nield (2008) argues, the primary purpose of science journalism is entertainment, not education. Television journalists are therefore looking for a good story, and will often try to create a level of suspense. By preparing for this, scientists can not only take more control of their television performances, but also do so in a way that is more engaging.

## 6   Relationship between scientists and journalists

Although scientists see it as their duty to interact with journalists (Section 2), there is still a fragile relationship between the two arenas. Both groups have contrasting motivations and expectations of what the interaction should look like and what the end result should be (Peters, 2013; Nield, 2008). Some of the friction between scientists and journalists arises because of different expectations. For example, scientists consider solid scientific knowledge to be an important indicator of quality, while journalists may just be looking for an entertaining story (Nield, 2008), or anyone to confirm their own suspicions or opinions, even if the topic goes beyond the scientists' expertise (Dijkstra et al., 2015). In the example below, Stoof experienced this latter type of friction.

### 6.1   Case: 'EénVandaag' interview on current news topic

On 20 Apr 2010, a heathland fire burned 60-80 ha near Hoog Soeren, the Netherlands, after a series of other fires occurred in the region. The day after, Stoof was interviewed for the news program

'EénVandaag'[6] who were interested in what could be done about these (for Dutch standards) frequent wildland fires. In the 5-minute item (Televisie Radio Omroep Stichting, 2010), video material of the fires was shown and two police and fire service professionals were also interviewed. During the interview, Stoof was asked who caused the fire problem in the Netherlands: the fire services or the land managers. Stoof is a soil scientist/hydrologist and while she was well-informed about fire issues in the Netherlands, she had not done research on this topic and was therefore not able to share any hard facts with the journalists. She therefore (evasively) responded that fire risk is something that needs to be taken into account when designing landscapes, and that fire services and nature organisations/land managers should enter a dialogue to reach an effective solution. Multiple takes were needed in which Stoof was asked to answer the same question over and over. In between takes the reporter asked her to phrase it "stronger, stronger". Stoof: "the interview was very uncomfortable because I was clearly not giving him what he wanted, but afterwards I was actually relieved that I did not give in, and realized I would rather not be on TV than be on TV and blame organisations (that I was hoping to collaborate with) based on no facts".

## 6.2 Relevant science communication literature and reflection

Stoof's story illustrates how the relationship between scientists and journalists can be rather delicate. As described in Section 2, scientists are interested and feel responsible to discuss their research with journalists. However, expectations and conditions are often different for the two groups (Peters, 2013; Dijkstra et al., 2015; Nield, 2008). For example, when discussing recent research findings, scientists find it very important that data are published in a scientific journal before they talk about it in the media. In addition, just like Stoof, most scientists are hesitant to discuss topics outside of what they consider to be their expertise or to make claims that they cannot back up with their own findings (Peters, 2013). In contrast, journalists seek out scientists as sources of knowledge not just on their own specific research topics, but also outside of their narrow area of expertise (Nelkin, 1995; Allgaier, 2011). Albaek (2011) observed that the role of journalists has shifted from merely reporting the news to interpreting what is going on in the world around us. Consequently, the role of scientists in the media has shifted from discussing their own research findings towards serving as referees or critics explaining and interpreting complex issues for society. In fact, Wien (2014) determined that only a fraction (17%) of Danish printed news items that included a scientist as expert were reporting specific research findings and in the majority of the news items, the role of the scientist was to comment on news events. This trend has been evolving over some time (Wien, 2014), and is highlighted here to make scientists aware of this development. At the same time, the way journalists choose which expert to approach is frequently influenced by rather subjective reasons such as their own interest in the field, the visibility of certain scientists in the media, personal

---

[6]Channel One Today

acquaintance with scientists, communication skills of the scientists, and their previous experiences with specific scientists (Allgaier, 2011).

The fact that journalists need scientists as experts to interpret the news and the subjective reasons for selecting specific scientists for that role lead to scientists being asked to comment on topics outside of what they feel is their field of expertise. Sometimes, journalists already know what they

want the scientist to say. In a Danish study, Albaek (2011) determined that in half of all interactions between scientists and journalists, the journalists only needed the scientist to confirm the message they wanted to express. As Stoof experienced, this can lead to uncomfortable situations.

Albaek (2011) proposes to find common ground in a time when current events are so complex that no single, unambiguous, scientifically proven solution exists. He states that scientists should feel an

375 obligation to participate in the public debate not as researchers who have the definite answer, but as "more or less insightful individuals" who give an expert assessment of current events. At the same time, journalists have the responsibility to make sure scientists' statements are indeed expert, instead of personal, opinions. When scientists understand a little better what journalists are looking for and when journalists are more aware of the reluctance and hesitation of scientists, the science-media

relationship can only improve (Dijkstra et al., 2015).

Even among journalists, differences exist between general journalists and science journalists, with the latter group being specialised in science writing/reporting and often having a background in science. Dunwoody (2004) states that training in science helps journalists become sensitive to scientific language and to evaluate evidence. She illustrates this with a story about three former players of a

385 professional football team in the United States who were diagnosed with ALS (Amyotrophic Lateral Sclerosis, a rapid and devastating neurodegenerative disease). Journalists without training in the scientific method and probability theory were quick to implicate heavy metals in the sewage sludge used to fertilize the sports field as a likely cause for the clustering of ALS cases, while trained science writers emphasized the lack of evidence and pointed to chance as the likely explanation.

Differences in journalists' expertise and background have important implications for geoscientists communicating their work, particularly when discussing abstract concepts. Risk and uncertainty are essential concepts in the planning of prevention and mitigation efforts of climate change as well as natural hazards, like floods, fires, landslides and earthquakes. Without careful communication and objective reporting, uncertainty can be misinterpreted but also politicized to downplay risks

(Liverman, 2008; Boykoff, 2008; Ward, 2008). Because of their greater familiarity with the scientific world, abstract concepts like risk and uncertainty are more easily conveyed through specialized science writers/reporters than through journalists without scientific training. Journalist background thereby shapes a science communication approach similar to the way that target audience shapes science communication approaches.

In summary, differences in expectations and motivations between scientists and journalists can sometimes cause friction. Where journalists may be looking for an expert to confirm their own con-

clusions, scientists often feel uncomfortable speaking outside of what they consider to be their field of expertise. Among journalists, differences exist in the amount of scientific training they have attained, leading to different levels of understanding about scientific topics, the scientific method, and

405 concepts such as risk and uncertainty. With a better understanding of both sides, scientists could serve as knowledgeable experts who can help interpret complicated current events. Although Wien (2014) suggests that relationships between journalists and scientists have improved, it would be interesting to study how well scientists are aware of the role they are being asked to play as interpreters and commentators.Possibly, this task is especially important for geoscience topics with high soci-

410 etal and political impact such as zoning for natural hazard prevention, hydrofracking for shale gas exploration, fire as a management tool, and climate change.

## 7 Stereotypes of scientists

In 1957 Mead and Métraux asked 35,000 high school students to describe a scientist (Mead and Metraux, 1957). The shared image that arose was "a man who wears a white coat and works in a

415 laboratory. He is elderly or middle aged and wears glasses". More than fifty years later this image still persists as repeated Draw-a-scientist-experiments show (Chambers, 1983; Finson, 2002). While these drawings necessarily produce only caricatures, the media seem all too happy to preserve (parts of) this stereotype. It was for instance a bit of a shock to the staff of the Guardian newspaper when science tv-show presenter Dara Ó Briain refused to wear a lab coat for a photo shoot: "Everyone

loved this idea except, crucially, Ó Briain." (Lewis, 2013). Stoof had a similar experience.

### 7.1 Case: 'Hoe?Zo!' science quiz

Hoe?Zo![7] was a popular science quiz on Dutch national TV in which two famous Dutch nationals answered questions posed by a team of three senior scientists, usually full professors. The quiz questions were sourced from a range of other scientists that are named during the show but who

are usually not present. They are typically illustrated with a sketch or simple experiment (Teleac). Stoof was asked to design a quiz question with an accompanying experiment. The answer to her question "What is a disaster related to forest fires?", being floods, was illustrated using two bench-scale rainfall simulators over wettable (non-hydrophobic) and water repellent (hydrophobic) soil. As fire can make soils water repellent which limits or prevents infiltration (Jordán et al., 2013), this

simple setup illustrated the increased risk of flooding from burned soils. On the day of production, Stoof (who was a PhD student at the time) was unexpectedly informed she had to stay behind the scenes. But because the experiment was 'complicated' she was later asked to conduct the experiment while a senior scientist explained the science. While the senior scientists wore normal clothes, Stoof (who hardly ever wears a lab coat and has a strong -negative- opinion about scientists wearing non-

---

[7]"Hoe?Zo!" translates to "How? Like this!", at the same time "hoezo?" translates to "why?"

functional lab coats on TV) was asked to put on a lab coat. When she resisted, a group of five people congregated to convince her, falsely telling her it was 'better for the light'. As she was about to give in to the pressure, the editor-in-chief assured her that if she really did not want to wear a lab coat on TV, she would not be forced to. The item was recorded without a lab coat.

### 7.2 Relevant science communication literature and reflection

In the show Stoof is wearing neutral clothes, but she is portrayed as assistant to the 'expert' of the show: an older male professor (who is is not a geoscientist, but a stem cell researcher). In this episode there is an expert panel, that explains all the topics, and a male host. Only one of the experts is female and they are all Caucasian. This is in line with how scientists are usually portrayed on television. In American prime-time dramatic programs 42% of the characters are female, but amongst portrayed

scientists only 30% are female. People of color are also underrepresented among scientists (Dudo et al., 2011). It seems that children's science education television programs in the United States are making an effort to steer away from stereotypes, but male characters still outnumber the female characters and minorities are less likely to be labeled scientists (Thayer and G, 2001). The Dutch lecture series Universiteit van Nederland (see section 3.1 and 5.1) currently has 66 episodes (in

November 2015), with only 6 being presented by women, a mere 9% . Only one presenter is non-Caucasian, a meager 1.5%. So the typical scientific role model on television is a white man. Women are less confident about their abilities in science, technology, engineering, and mathematics (STEM) fields when they are confronted with stereotypes of scientists. Although Cheryan et al. (2011) show that the gender of role models is not the biggest factor in undermining women's scientific self esteem:

it is the general stereotype of the nerd. As such, stereotypical role models with glasses, a shirt with 'I code therefore I am', socks and sandals and hobbies such as gaming and science fiction have a negative effect on the confidence of women in their STEM-abilities. As geoscience is part of STEM, these findings are also relevant for the geosciences. Therefore it is important to offer a wider range of role-models of scientists on TV. Research by Nosek et al. (2009) shows that this is also relevant in the

context of enthusing girls for science. Worldwide, boys outperform girls in science in the 8-th grade, but the size of this gender gap varies per country. Nosek et al. (2009) discovered that nation-level sex differences in 8th-grade science and mathematics achievement are predicted by implicit stereotypes. This was illustrated by a large-scale 34-country survey using Implicit Association Tests (IAT) to measure the strength of a person's automatic associations between different concepts. In over half

a million IAT's, 70% of people implicitly associated science more with males than with females. Nosek et al. (2009) argue that these stereotypes have a causal effect on inequalities in science and that the lack of diversity in science reinforces stereotypical images.

In conclusion, avoiding stereotypes of scientists in geoscience communication on TV is beneficial for participation of minorities in this field. There is no data of the degree to which minorities com-

470 municate the geosciences, but given that geoscience itself is male-dominated (Holmes et al 2008),

it is likely that there are more male than female geoscientists on TV. When more female and non-caucasian researchers act as experts on television, minorities may be encouraged to choose a career in science, potentially leading to a bigger and more diverse pool of future geoscientists.

## 8   Conclusions

To provide background knowledge for geoscience communicators, we reviewed theory of science communication through television in the context of geosciences. We discussed six major themes: scientist motivation (Section 2), target audience (Section 3), jargon and information transfer (Section 4), narratives and storytelling (Section 5), relationship between scientists and journalists (Section 6), stereotypes of scientists (Section 7). While much of the science communication literature is directed

towards other fields (for instance medical sciences in the case of jargon), most general concepts are also applicable to the geosciences. There is for instance great potential for using narrative and storytelling in this field, because natural hazards and other geoscience topics often have a human element: that of risk and health (Stewart and Nield, 2013). And because of contentious geoscientific topics like preparation for and probability of hazards and risks, potential friction between scientists

and journalists needs to be taken into account when preparing for science communication activities. This friction is not always over content but because of persistent stereotypes of scientists on TV, friction may also occur over appearances. Standing up against the white male in a lab coat stereotype to change the face of science should not be a challenge for geoscientists, where female participation is steadily increasing (Holmes et al., 2008; American Geosciences Institute., 2014) and whose work

is often outdoors in very interesting landscapes around the world. Jargon on the other hand can be a challenge, as it can be difficult to recognize what constitutes jargon in the geosciences where researchers may not be aware that they are using complex vocabulary. Further research on jargon in the geosciences may greatly contribute to geoscience science communication activities that are accessible for all, including for those with limited literacy that make up a large part of the general

public. TV is a very powerful medium and offers a unique opportunity to reach many viewers. Conveying even small amounts of information to such large audiences can engage large numbers of people in the geosciences and enthuse them for it. Increased understanding of effective geoscience communication on TV can thereby positively contribute to the geoscience debate in society.

## 9   Author contributions

All authors contributed equally. R.W. Hut and C.R. Stoof described case studies, placed findings in the geoscience context, A.M. Land-Zandstra and I. Smeets reviewed science communication literature and analyzed the video material and placed this in the context of the reviewed literature.

## Appendix A

### A1

*Acknowledgements.* We gracefully acknowledge the broadcast organizations AVROTROS, BNNVARA, NTR (TELEAC), and KRO-NRCV for granting permission to subtitle and make available the TV clips discussed in this paper, as well as Nes Emin Oglou, Christel van der Pijl and Hank Willemers from the Institute of Sound and Vision for their kind help in arranging this permission. In addition, R.W. Hut and C.R. Stoof would like to thank a number of people for facilitating our science communication activities and/or providing support

and discussion: Roy Meijer, Diederik Jekel, the UvNL crew, Jac Niessen, Bouke de Vos, Desiree Hoving, Peter Vermeulen and Coen Ritsema. We furthermore thank Ted Nield, Fergus McAuliffe and one anonymous reviewer for their valuable comments and anecdotes that improved an earlier version of this manuscript.

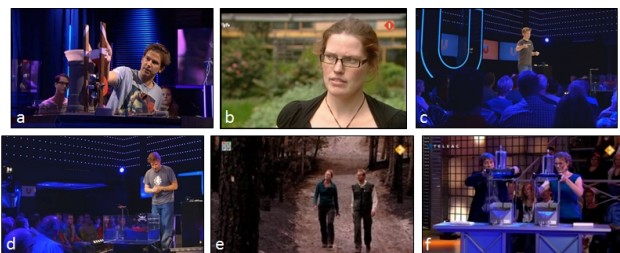

**Figure 1.** screenshots of the examples used in this article: a) 'De Wereld Leert Door' science talk show used in Section 2 on motivation. b) 'Eén Vandaag' news background show used in Section 6 on the relationship between scientists and journalists. c) 'Universiteit van Nederland' public lecture used in Section 3 on target audiences. d) 'Universiteit van Nederland' public lecture used in Section 5 on narratives and storytelling. e) 'Netwerk' news background show used in Section 4 on jargon and information transfer. f) 'Hoe?Zo!' science quiz used in Section 7 on stereotypes of scientists

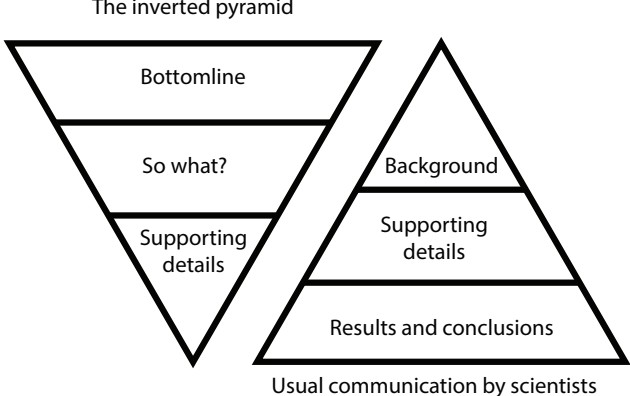

**Figure 2.** The inverted pyramid versus usual communication by scientists. In the inverted pyramid, conclusions (the bottomline) are presented first, followed by further details (narrowing down of the triangle). Scientific communications typically start with details (the background), then increasing in breadth towards the conclusions (widening of the triangle)

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

**Table 1.** Number of views on YouTube for the different Universiteit van Nederland lectures given by Hut. Where one normally would expect a declining number with lectures further in the series, the second lecture, that strongly uses narrative, has a higher number of views than the first lecture.

| Episode | Title | Number of views |
| --- | --- | --- |
| 1 | Why do you only need a postcard and a picture frame to become an inventor? | 8,743 |
| 2 | Why do boats keep on hitting bridges? | 10,400 |
| 3 | How can you recognize Beethoven's fifth with mathematics? | 9,242 |
| 4 | Why does scientific research not have to be expensive? | 5,345 |
| 5 | How can you use a garden hose to predict the weather? | 6,597 |