# Peer review of "Manuscript prepared for Hydrol. Earth Syst. Sci. with version 2015/09/17 7.94 Copernicus papers of the LATEX class copernicus.cls. Date: 7 April 2016"

_Hydrology and Earth System Sciences, 2015_

## Referee Comment (RC1) · T. Nield (Referee) · 21 Jan 2016

I found the paper informative and well supported by evidence and its discussion insightful. I would recommend some minor textual revisions.

Very trivially, on p5 the phrase 'as it is the most important source to learn about science' might be better [hrased - I suggest something along the lines of 'as it is the main source of science information for the general public'. A little lower down (line 15) 'that is' could be removed with benefit to the elegance of the sentence. In line 17 'to inform them of what happened with their money' is OK but a little gauche. Suggest 'towards the taxpayers who have a right to know what has been done with their money'.
[Figure]

Page 13, ln 10: 'phenomena' should read 'phenomenon'.

Here we come to the substantive issue of scientists' reluctance to 'speak outside their areas of expertise' - see line 13: 'outside their expertise'. I would suggest 'outside what they regard as their expertise' here, and ind eed throughout. The difficulty scientists have in this area is that they define their areas of expertise over-minutely. In fact, as far as the public is concerned, a scientist's area of expertise is a lot broader than they (the scientists) would think, and the public would be correct. When journalists ask scientists to comment 'more broadly' they are asking for the opinion of someone who is vastly more qualified in (probably) ALL areas of 'science' than 99.9% of everyone watching/listening. The problem here is scientists' habit of only ever mixing with other scientists - it gives them a warped view of what expertise means in the real world. And it impinges on their reluctance to tread on the toes of others ('Oh, you should speak to Professor Dingbat on THAT topic, not me'). Professor Dingbat isn't there, and can't be found in time. You are there, and your opinion might not be up to Dingbat standards of authority but as near as makes no difference when seen from afar.

I am not generally impressed by 'draw-a-scientist' experiments, as cited in line 22. When people are asked to 'draw a scientist' they interpret this to mean 'draw a cari-cature of a scientist', and they oblige. The question is interpreted as 'make a drawing I will recognise as representing a scientist'. Most people's graphic skills are not up to doing much else, anyway. If you ask anyone today to 'draw a railway train', most of them will draw a steam engine at the front, irrespective of whether they have ever actually seen one. Almost nobody in society ever knowingly meets 'a scientist' - they are too few. There are more members of Badminton Racquets Clubs in the UK than there are people with scientific qualifications, and yet most people do not know anyone who belongs to one. It is hardly surprising that stereotypes persist in this information

vacuum.

The story about Dara O'Briain (and others) being asked to conform to stereotypes and refusing (ie refusing to don a white labcoat) is interesting, because I have an anecdote in this context of stereotyping and identification of 'who has the expertise'. A BBC geology TV series broadcast in the 1990s featured a team of presenters, led by a well known 'talent' who however had no special knowledge of geology. Among the subordinate presenters (each of whom did a segment or two linked by the talent) was a qualified geologist with extensive communications experience as a journalist. She also acted as (unpaid, unofficial) scientific script editor for the entire series.

However, when it came to taking the 'team photo' for Radio Times in front of the Old Man of Hoy, the photographer asked the female geologist to 'give the hammer' she was holding to the talent, standing centrally. This she resolutely refused to do, because she regarded herself as the only qualified geologist on the show, and the hammer was HER badge of office not to be usurped!

---

## Referee Comment (RC2) · Anonymous Referee #2 · 26 Jan 2016

This interesting, well-written paper represents a solid attempt on the highly relevant research field of geoscience communication. It focuses on geo-communicators in the field of natural hazards that use television as a medium to disseminate their messages. Despite the title, the papers findings are not limited to geoscience on television, but applicable to geocommunication in a wider sense. The authors claim that an awareness for important findings of science communication research is relatively low among geoscientists. They consequently review science communication literature and consider theoretical concepts relevant for approaching six major themes of geoscience communication: scientist motivation, target audience, narratives and storytelling, jargon and information transfer, relationship between scientists and journalists, and stereotypes of

scientists on TV. Each theme is illustrated by a case study of geosciences on TV. The authors link the case studies to the findings of their theoretical research and critically discuss implications for geoscientists. This methodological approach is valid and interesting and the case studies in combination with the theoretical input are informative. Yet, there are three aspects that might be worth taking a closer look at, to make sure the scientific up-to-dateness of the paper is given.

1. In the introduction of the paper on page 2 it is stated that "effective communication of hydrogeoscience information to the general public is becoming increasingly important." Here it might be worth clarifying how effectiveness is understood and what the features of an effective science communication are. Do the authors primordially focus on an increase of public awareness and a bigger scientific literacy? Or does their concept also encompass the public understanding of science and public participation? Especially in the field of natural hazard communication it is crucial to make sure that communication genuinely helps people at risk. Risk communication studies point out only little or no correlation between a heightened scientific literacy / risk awareness and risk adaptation measures. This provokes the question if communication can be considered as "effective" when it does not contribute to an increased preparedness. Therefore I would recommend a more holistic approach towards the idea of an "effective communication" that is not reduced to effective "knowledge transfer".

2. In section 3 the authors address the theme of "target groups" (page 6-9). The authors stress the necessity to think more in-depth about possible audiences and their requirements. Although this reflection is highly relevant, the concept of target groups lacks some differentiation. For example, the authors state that "it is extremely difficult – if not impossible – to successfully reach more than one target audience" (page 6) without having defined what exactly constitutes a target group. They also describe "the general public" as a target group (page 8), which can be easily seen as a contradiction to their initial claim for a more differentiated approach.

3. The paper provokes the question of how geoscientist shall actually better understand target groups. It is widely shown that a one-way-mode of communication where the "knowing scientists" explains "scientific facts" to a lay audience (deficit model) has limitations. In order to avoid working with preconceptualized images of target groups or worth "the general public", it would be useful also to stress more in detail the role of interactivity in the design of geocommunication. A reflection on "narrative and storytelling" and the "avoidance of jargons" as in section 4 and 5 of the paper are important to better reach audiences, but are not sufficient to address major shortcomings of science communication. The role of interaction with target groups and negotiation of mutual understandings are neglected in the paper and might be worth taken into account. Linking a case study with findings that indicate that one-way communication on television has not only benefits, as outlined on page 6 and 11, but also major restrictions, would be a useful to broaden the scope of the paper. Especially in empirical social science research there is a growing body of literature on the role of knowledge on social and cultural conditions which can turn out to be crucial elements to consider in the training and design of geocommunication.

---

## Referee Comment (RC3) · F. McAuliffe (Referee) · 9 Feb 2016

General comments

This is well written and informative manuscript. The six themes outlined are well discussed in the context of television using the method of opening description, case study, and reflection. This consistent structure aids to understanding. Some minor revisions are recommended.

Specific comments

Page 12 – line 6: use of the word "outreach". In my experience, this word, has come to mean "teaching children about science" in the eyes of some scientists/geoscientists.

For this reason, I have moved away from using this term, and moved towards "engagement" which is a better catch all term for connecting with many audiences, from children up to retirees. Suggest changing "outreach" to "engagement activities/efforts/initiatives" throughout.

Page 12 – line 6: "intended audience": having covered target audiences in section 3 it might be more appropriate to stick with this terminology in the rest of the article. Suggested change from "intended audience" to "target audience". Also agree with the comment of Anonymous Referee #2 about the need for a definition of target groups to be included in section 3. Including a definition here will aid understanding when target groups are referred to later in the article.

Page 12 – lines 19-21: you have mentioned that the difference in pyramid layout between the convention and the Stewart and Nield (2013) paper is interesting. Consequently, why it is interesting needs to be explained here.

Page 16 – the idea of the role of the scientist in the media changing from discussing their own research to interpreting complex issues for society is an important idea. This is a big barrier to getting scientists to engage in non-scripted media interactions, for fear of being asked to comment on something outside of their own area of expertise (which is in fact becoming increasingly more refined). Some scientists now feel that they must have published (extensively) on a very particular topic, before they will even attempt to answer media questions about it. A nice link is made in this section between the work of Albeak (2011) and the consequences of this for scientists. While the point is made that the role of scientists in the media has changed, can this point be linked to any literature on whether scientists are actually aware of this change?

Page 18 – line 26: "Something similar happened to Stoof". This is a bit blunt. Suggest changing to "Stoof had a similar experience".

Page 19 - line 2 – When was Hoe?Zo! on Dutch national TV? It would be good to have a date here for context.

Page 30 – Figure 2: the use of parenthesis is somewhat confusing here. Use of "(the bottomline)", where the parenthesis is used to direct readers' attention to a specific layer of the pyramid is fine. However, the second use of parenthesis "(narrowing down of the triangle)" caused me to initially try to find the layer labelled as "narrowing down of the triangle" within the pyramid. Suggest changing the second sentence to "In the inverted pyramid, conclusions (the bottomline) are presented first, followed by further details i.e. narrowing down of the pyramid." This suggested change can also be reflected in the third sentence. Avoid inter-changing "pyramid" and "triangle".

Technical corrections

Page 13 – line 23: change from "Pulitzer-prize winning" to "Pulitzer prize-winning"

Page 15 – lines 8 and 9: "someone" is singular, so it is likely that it is a singular scientist that is being referred to in line 9:

Page 16 – line 12: change from "data is published" to "data are published"

Page 18 – line 21: change from "this image still persist as" to "this image still persists as"

Page 18 – line 24: change from "newspaper the Guardian" to "the Guardian newspaper"

Page 20 - line 4: change from "only 30% is female" to "only 30% are female" as earlier in the same sentence "42%" is followed by "are".

Page 21 – lines 16-21. Sentence is too long and needs to be broken. Suggest breaking after "applicable to the geosciences".

Page 22 – line 14: change from "little information" to "small amounts of information"

Page 22 – line 23: change from "KRO-NRCVfor" to "KRO-NRCV for"

Page 28 – Table 1: change from "Number of views on Youtube" to "Number of views

on YouTube".

---

## Author Comment (AC1) · 14 Mar 2016

We thank the reviewers for their thorough reading of our work, their constructive criticism and their suggestions that, in our opinion, helped improve the quality and relevance of our manuscript. We specifically like to thank the reviewers for their kind words regarding the thoroughness of our literature review and the general usefulness of our manuscript to the readership of HESS.

*Our replies to the reviewers comments are provided in italics below.*

RC1

T. Nield (Referee)

[Figure]

I found the paper informative and well supported by evidence and its discussion insightful. I would recommend some minor textual revisions.

*We thank dr. Nield for these kind words.*

Very trivially, on p5 the phrase 'as it is the most important source to learn about science' might be better phrased - I suggest something along the lines of 'as it is the main source of science information for the general public'.

*This was changed as per dr. Nields suggestion.*

A little lower down (line 15) 'that is' could be removed with benefit to the elegance of the sentence.

*This was changed as per dr. Nields suggestion.*

In line 17 'to inform them of what happened with their money' is OK but a little gauche. Suggest 'towards the taxpayers who have a right to know what has been done with their money'.

*This was changed as per dr. Nields suggestion.*

Page 13, ln 10: 'phenomena' should read 'phenomenon'.

*This was changed as per dr. Nields suggestion.*

Page 16 Here we come to the substantive issue of scientists' reluctance to 'speak outside their areas of expertise' - see line 13: 'outside their expertise'. I would suggest 'outside what they regard as their expertise' here, and indeed throughout. The difficulty scientists have in this area is that they define their areas of expertise over-minutely. In fact, as far as the public is concerned, a scientist's area of expertise is a lot broader than they (the scientists) would think, and the public would be correct. When journalists ask scientists to comment 'more broadly' they are asking for the opinion of someone who is vastly more qualified in (probably) ALL areas of 'science' than 99.9% of everyone watching/listening. The problem here is scientists' habit of only ever mixing with other

scientists - it gives them a warped view of what expertise means in the real world. And it impinges on their reluctance to tread on the toes of others ('Oh, you should speak to Professor Dingbat on THAT topic, not me'). Professor Dingbat isn't there, and can't be found in time. You are there, and your opinion might not be up to Dingbat standards of authority but as near as makes no difference when seen from afar.

*We have changed the wording throughout to emphasize what scientists consider their field of expertise. We also have clarified the paragraphs about the topic of expertise to include this difference in what the media expect from scientists and what scientists consider appropriate to comment about. Lines 335-340, which used to read: "Consequently, the role of scientists in the media has shifted from discussing their own research findings towards serving as referees or critics explaining and interpreting complex issues for society. At the same time, the way journalists choose which expert to approach is frequently influenced by rather subjective reasons such as their own interest in the field, the visibility of certain scientists in the media, personal acquaintance with scientists, communication skills of the scientists, and their previous experiences with specific scientists (Allgaier, 2011)." have been changed to: "Consequently, the role of scientists in the media has shifted from discussing their own research findings towards serving as referees or critics explaining and interpreting complex issues for society. In fact, Wien (2014) determined that only a fraction (17%) of Danish printed news items that included a scientist as expert were reporting specific research findings and in the majority of the news items, the role of the scientist was to comment on news events. This trend has been evolving over some time (Wien, 2014), and is highlighted here to make scientists aware of this development. At the same time, the way journalists choose which expert to approach is frequently influenced by rather subjective reasons such as their own interest in the field, the visibility of certain scientists in the media, personal acquaintance with scientists, communication skills of the scientists, and their previous experiences with specific scientists (Allgaier, 2011)." (lines 343-358)*

Page 18 I am not generally impressed by 'draw-a-scientist' experiments, as cited in

line 22. When people are asked to 'draw a scientist' they interpret this to mean 'draw a caricature of a scientist', and they oblige. The question is interpreted as 'make a drawing I will recognise as representing a scientist'. Most people's graphic skills are not up to doing much else, anyway. If you ask anyone today to 'draw a railway train', most of them will draw a steam engine at the front, irrespective of whether they have ever actually seen one. Almost nobody in society ever knowingly meets 'a scientist' - they are too few. There are more members of Badminton Racquets Clubs in the UK than there are people with scientific qualifications, and yet most people do not know anyone who belongs to one. It is hardly surprising that stereotypes persist in this information vacuum.

*We agree that the drawings are necessarily caricatures and added a sentence about this. " The media seem all too happy to preserve (parts of) this stereotype." Has been changed to "While these drawings necessarily produce only caricatures, the media seem all too happy to preserve (parts of) this stereotype." (line 410)*

The story about Dara O'Briain (and others) being asked to conform to stereotypes and refusing (ie refusing to don a white labcoat) is interesting, because I have an anecdote in this context of stereotyping and identification of 'who has the expertise'. A BBC geology TV series broadcast in the 1990s featured a team of presenters, led by a well known 'talent' who however had no special knowledge of geology. Among the subordinate presenters (each of whom did a segment or two linked by the talent) was a qualified geologist with extensive communications experience as a journalist. She also acted as (unpaid, unofficial) scientific script editor for the entire series. However, when it came to taking the 'team photo' for Radio Times in front of the Old Man of Hoy, the photographer asked the female geologist to 'give the hammer' she was holding to the talent, standing centrally. This she resolutely refused to do, because she regarded herself as the only qualified geologist on the show, and the hammer was HER badge of office not to be usurped!

*We thank dr. Nields for this interesting anecdote. Since we used the Dara O'Briain*

*anecdote as an illustration to our argument, and the article is running long as it is, we have decided not to include another similar anecdote. We do like to ask dr. Nields if it is ok, with proper attribution, to use his anecdote in presentations we give on this subject.*

RC2

anonymous

This interesting, well-written paper represents a solid attempt on the highly relevant research field of geoscience communication. It focuses on geo-communicators in the field of natural hazards that use television as a medium to disseminate their messages. Despite the title, the papers findings are not limited to geoscience on television, but applicable to geocommunication in a wider sense. The authors claim that an awareness for important findings of science communication research is relatively low among geoscientists. They consequently review science communication literature and consider theoretical concepts relevant for approaching six major themes of geoscience communication: scientist motivation, target audience, narratives and storytelling, jargon and information transfer, relationship between scientists and journalists, and stereotypes of scientists on TV. Each theme is illustrated by a case study of geosciences on TV. The authors link the case studies to the findings of their theoretical research and critically discuss implications for geoscientists. This methodological approach is valid and interesting and the case studies in combination with the theoretical input are informative.

*We thank the reviewer for these kind words.*

Yet, there are three aspects that might be worth taking a closer look at, to make sure the scientific up-to-dateness of the paper is given.

1. In the introduction of the paper on page 2 it is stated that "effective communication of hydrogeoscience information to the general public is becoming increasingly important." Here it might be worth clarifying how effectiveness is understood and what the features

of an effective science communication are. Do the authors primordially focus on an increase of public awareness and a bigger scientific literacy? Or does their concept also encompass the public understanding of science and public participation? Especially in the field of natural hazard communication it is crucial to make sure that communication genuinely helps people at risk. Risk communication studies point out only little or no correlation between a heightened scientific literacy / risk awareness and risk adaptation measures. This provokes the question if communication can be considered as "effective" when it does not contribute to an increased preparedness. Therefore I would recommend a more holistic approach towards the idea of an "effective communication" that is not reduced to effective "knowledge transfer".

*In this paper we focus on increase of public awareness and bigger scientific literacy, since television is a one-way medium. We added a paragraph to clarify this choice and emphasize that two-way communication and engagement are important too, especially in hazard situations (note that none of our examples was about a hazard situation). The paragraph we added reads: "Television is by definition a form of one-way communication, and in this context effective science communication is aimed at popularizing science and enhancing the scientific literacy of the general public. However, within the field of science communication this top-down-approach has been criticized (Sturgis and Allum, 2004). Two-way communication is important, for instance since scientific literacy is not the only factor in how people perceive climate change and other risks (Kahan et al., 2012). In hazard situations it is especially important to engage the public instead of just informing them (Frewer, 2004). However, this will be easier if there is a broader "geo-literacy" among the general public (Stewart and Nield, 2013) and despite its limitations one-way communication has its own merits (Sturgis and Allum, 2004),(Wright, 2006). Recent results also show that the general public values one-way communication by experts, indicating that the public is looking for new knowledge and understanding of science (Fogg-Rogers et al., 2015)." (lines 30-40)*

2. In section 3 the authors address the theme of "target groups" (page 6-9). The

authors stress the necessity to think more in-depth about possible audiences and their requirements. Although this reflection is highly relevant, the concept of target groups lacks some differentiation. For example, the authors state that "it is extremely difficult if not impossible to successfully reach more than one target audience" (page 6) without having defined what exactly constitutes a target group. They also describe "the general public" as a target group (page 8), which can be easily seen as a contradiction to their initial claim for a more differentiated approach.

*We have included a more specific description of what we mean with general public (i.e. a general audience without relevant scientific background) as opposed to the students who do have relevant background. We have also clarified what we mean by target audience. "The target audience for UvNL is the general public." was changed to "The target audience for UvNL is a general audience without specific scientific background." (line 150) and this was consequently changed throughout the manuscript.*

3. The paper provokes the question of how geoscientist shall actually better understand target groups. It is widely shown that a one-way-mode of communication where the "knowing scientists" explains "scientific facts" to a lay audience (deficit model) has limitations. In order to avoid working with preconceptualized images of target groups or worth "the general public", it would be useful also to stress more in detail the role of interactivity in the design of geocommunication. A reflection on "narrative and storytelling" and the "avoidance of jargons" as in section 4 and 5 of the paper are important to better reach audiences, but are not sufficient to address major shortcomings of science communication. The role of interaction with target groups and negotiation of mutual understandings are neglected in the paper and might be worth taken into account. Linking a case study with findings that indicate that one-way communication on television has not only benefits, as outlined on page 6 and 11, but also major restrictions, would be a useful to broaden the scope of the paper. Especially in empirical social science research there is a growing body of literature on the role of knowledge on social and cultural conditions which can turn out to be crucial elements to consider

in the training and design of geocommunication.

*We did not discuss interaction with target groups, since the focus of this paper is on television. We agree that one-way-communication and the deficit model have their restrictions and added references to two papers that discuss the role of the deficit model. We also supplied a reference to a recent study about the public preference for one-way communication from experts to show that this form of communication is still in demand. We added: "Television is by definition a form of one-way communication, and in this context effective science communication is aimed at popularizing science and enhancing the scientific literacy of the general public. However, within the field of science communication this top-down-approach has been criticized (Sturgis and Allum, 2004). Two-way communication is important, for instance since scientific literacy is not the only factor in how people perceive climate change and other risks (Kahan et al., 2012). In hazard situations it is especially important to engage the public instead of just informing them (Frewer, 2004). However, this will be easier if there is a broader "geo-literacy" among the general public (Stewart and Nield, 2013) and despite its limitations one-way communication has its own merits (Sturgis and Allum, 2004),(Wright, 2006). Recent results also show that the general public values one-way communication by experts, indicating that the public is looking for new knowledge and understanding of science (Fogg-Rogers et al., 2015)." (lines 30-40)*

RC3

F. McAuliffe (Referee)

General comments This is well written and informative manuscript. The six themes outlined are well discussed in the context of television using the method of opening description, case study, and reflection. This consistent structure aids to understanding. Some minor revisions are recommended.

*We thank Mr McAuliffe for these kind words.*

Specific comments

Page 12 line 6: use of the word "outreach". In my experience, this word, has come to mean "teaching children about science" in the eyes of some scientists/geoscientists. For this reason, I have moved away from using this term, and moved towards "engagement" which is a better catch all term for connecting with many audiences, from children up to retirees. Suggest changing "outreach" to "engagement activities/ efforts/initiatives" throughout.

*We agree with Mr McAuliffe and changed "outreach" to "science communication activities" throughout the manuscript.*

Page 12 line 6: "intended audience": having covered target audiences in section 3 it might be more appropriate to stick with this terminology in the rest of the article. Suggested change from "intended audience" to "target audience". Also agree with the comment of Anonymous Referee 2 about the need for a definition of target groups to be included in section 3. Including a definition here will aid understanding when target groups are referred to later in the article.

*We agree with Mr McAuliffe and have changed the wording to "target audience" throughout the paper. See the response to Referee 2 about definition of target groups.*

Page 12 lines 19-21: you have mentioned that the difference in pyramid layout between the convention and the Stewart and Nield (2013) paper is interesting. Consequently, why it is interesting needs to be explained here.

*We agree with Mr McAuliffe that using the word "interesting" puts a burden on us to explain why it is interesting. This was not our intention, we were merely surprised by the deviation from the standard in the science communication literature. Accordingly, we changed "interesting" to "surprisingly". The sentence now reads: "Surprisingly to us, they showed the inverted pyramid as a normal (upright) pyramid, where the convention in the science communication literature is to depict it as in (Fig 2, right pyramid)." (lines*

[Figure]

*261-263)*

Page 16 the idea of the role of the scientist in the media changing from discussing their own research to interpreting complex issues for society is an important idea. This is a big barrier to getting scientists to engage in non-scripted media interactions, for fear of being asked to comment on something outside of their own area of expertise (which is in fact becoming increasingly more refined). Some scientists now feel that they must have published (extensively) on a very particular topic, before they will even attempt to answer media questions about it. A nice link is made in this section between the work of Albeak (2011) and the consequences of this for scientists. While the point is made that the role of scientists in the media has changed, can this point be linked to any literature on whether scientists are actually aware of this change?

*See also response to dr. Nields comment about this same section. We have clarified this difference between what the media expect and what scientists find appropriate to comment about. We could not find any literature about the awareness of scientists about this difference. Therefore we have included this as an interesting topic for future research. We furthermore added: "Although Wien (2014) suggests that relationships between journalists and scientists have improved, it would be interesting to study how well scientists are aware of the role they are being asked to play as interpreters and commentators.Possibly, this task is especially important for geoscience topics with high societal and political impact such as zoning for natural hazard prevention, hydrofracking for shale gas exploration, fire as a management tool, and climate change." (lines 398-404)*

Page 18 line 26: "Something similar happened to Stoof". This is a bit blunt. Suggest changing to "Stoof had a similar experience".

*This was changed as per Mr McAuliffe suggestion.*

Page 19 - line 2 When was Hoe?Zo! on Dutch national TV? It would be good to have a date here for context.

*Due to an error on our side, the year of publication (2009) was not mentioned in the BibTex file, this has been fixed.*

Page 30 Figure 2: the use of parenthesis is somewhat confusing here. Use of "(the bottomline)", where the parenthesis is used to direct readers' attention to a specific layer of the pyramid is fine. However, the second use of parenthesis "(narrowing down of the triangle)" caused me to initially try to find the layer labelled as "narrowing down of the triangle" within the pyramid. Suggest changing the second sentence to "In the inverted pyramid, conclusions (the bottomline) are presented first, followed by further details i.e. narrowing down of the pyramid." This suggested change can also be reflected in the third sentence. Avoid inter-changing "pyramid" and "triangle".

Technical corrections

Page 13 line 23: change from "Pulitzer-prize winning" to "Pulitzer prize-winning"

*This was changed as per mr. McAuliffe's suggestion.*

Page 15 lines 8 and 9: "someone" is singular, so it is likely that it is a singular scientist that is being referred to in line 9:

*"someone" was changed to "anyone".*

Page 16 line 12: change from "data is published" to "data are published"

*This was changed as per mr. McAuliffe's suggestion.*

Page 18 line 21: change from "this image still persist as" to "this image still persists as"

*This was changed as per mr. McAuliffe's suggestion.*

Page 18 line 24: change from "newspaper the Guardian" to "the Guardian newspaper"

*This was changed as per mr. McAuliffe's suggestion.*

Page 20 - line 4: change from "only 30% is female" to "only 30% are female" as earlier in the same sentence "42%" is followed by "are".

*This was changed as per mr. McAuliffe's suggestion.*

Page 21 lines 16-21. Sentence is too long and needs to be broken. Suggest breaking after "applicable to the geosciences".

*We removed the sentence, since it was repeating earlier statements and potentially creating confusion.*

Page 22 line 14: change from "little information" to "small amounts of information"

*This was changed as per mr. McAuliffe's suggestion.*

Page 22 line 23: change from "KRO-NRCVfor" to "KRO-NRCV for"

*This was changed as per mr. McAuliffe's suggestion.*

Page 28 Table 1: change from "Number of views on Youtube" to "Number of views on YouTube".

*This was changed as per mr. McAuliffe's suggestion.*

Please also note the supplement to this comment:
http://www.hydrol-earth-syst-sci-discuss.net/hess-2015-518/hess-2015-518-AC1-supplement.pdf

[Figure]

**Supplement:**

[revised manuscript text omitted]

---

## Author Response (AR1)

Manuscript prepared for Hydrol. Earth Syst. Sci.
with version 2015/09/17 7.94 Copernicus papers of the LaTeX class copernicus.cls.
Date: 7 April 2016

**Geoscience on television: a review of science communication literature in the context of geosciences**

Rolf Hut[1], Anne M. Land-Zandstra[2], Ionica Smeets[2], and Cathelijne Stoof[3]

[1]Delft University of Technology, Faculty of Civil Engineering and Geosciences, chair of Water Resources Engineering
[2]Science, Communication and Society, Leiden University
[3]Soil Geography and Landscape Group, Wageningen University

*Correspondence to:* Rolf Hut (r.w.hut@tudelft.nl)

**Abstract.** We thank the editor for his kind words, time and effort. We have incorporated his final comment. The paragraph now reads:

In academic interactions, whether on paper or at conferences or in seminars, conclusions are usually presented at the end (Fig. 2, right pyramid). This is in stark contrast to journalism where it is common to start with the conclusions ( Fig. 2, left pyramid). This latter structure is called the inverted pyramid (Pottker 2003): the most important information comes first and the lead sentence answers the five w-questions: who? when? where? what? why? The rest of the facts are then presented in decreasing order of importance. Stewart and Nield argue that geoscientists can communicate more effectively with laymen when they invert the pyramid of their presentations (Stewart and Nield, 2013). The paper that first introduced the inverted pyramid (Somerville and Hassol, 2011) to geoscientists uses a down pointing triangle for the scientists communicating with peers and an up pointing triangle for scientists communicating with lay people to convey the message that "the scientific method trains scientists to begin with outlining the general academic context and then work through to some narrow research finding, which they then proceed to communicate. The starting point for the public, on the other hand, is the sharp question of 'so what?', and addressing that successfully can take them deeper into the academic detail and wider context." (Stewart 2016). Here, we use the convention used by journalists, where the inverted pyramid of communicating with lay people is a down pointing triangle. We recognize that this is a matter of personal preference and at the same time, that the different conventions used in different fields to communicatie ideas like the inverted pyramid is a valid subject of academic study in its own right.